∂ | **Open Peer Review** | Bacteriology | Research Article

# Pentamidine inhibition of streptopain attenuates *Streptococcus pyogenes* virulence

Keya Trivedi,[1] Christopher N. LaRock[2,3,4]

**ABSTRACT** The obligate human pathogen *Streptococcus pyogenes* (also known as GAS; Group A *Streptococcus*) carries high morbidity and mortality, primarily in impoverished or resource-poor regions. The failure rate of monotherapy with conventional antibiotics is high, and invasive infections by this bacterium frequently require extensive supportive care and surgical intervention. Thus, it is important to find new compounds with adjunctive therapeutic benefits. The conserved secreted protease streptopain (Streptococcal pyogenic exotoxin B; SpeB) directly contributes to disease pathogenesis by inducing pathological inflammation, degrading tissue, and promoting the evasion of antimicrobial host defense proteins. This study screened 400 diverse off-patent drugs and drug-like compounds for inhibitors of streptopain proteolysis. Lead compounds were tested for activity at lower concentrations and anti-virulence activities during *in vitro* infection. Significant inhibition of streptopain was seen for pentamidine, an anti-protozoal drug approved for the treatment of Pneumocystis pneumonia, leishmaniasis, and trypanosomiasis. Streptopain inhibition rendered GAS susceptible to killing by human innate immune cells. These studies identify unexploited molecules as new starting points for drug discovery and a potential for repurposing existing drugs for the treatment of infections by GAS.

**IMPORTANCE** *Streptococcus pyogenes* is a common cause of severe invasive infections. Repeated infections can trigger autoimmune diseases such as acute rheumatic fever and rheumatic heart disease. This study examines how targeting a specific, highly conserved virulence factor of the secreted cysteine protease streptopain can sensitize a serious pathogen to killing by the immune system. Manipulating the host-pathogen interaction, rather than attempting to directly kill a microbe, is a promising therapeutic strategy. Notably, its benefits include limiting off-target effects on the microbiota. Streptopain inhibitors, including the antifungal and antiparasitic drug pentamidine as identified in this work, may therefore be useful in the treatment of *S. pyogenes* infection.

**KEYWORDS** *Streptococcus pyogenes*, protease, virulence factor, antibiotics, anti-infective, drug repurposing

*S*treptococcus pyogenes (also known as GAS; Group A *Streptococcus*) is a top cause of infectious mortality and is responsible for over half a million annual deaths worldwide (1). Humans are colonized throughout childhood, leading to bouts of pharyngitis ("strep throat") or pyoderma, but often without overt symptoms of disease (2). Rheumatic heart disease can follow repeated infections, while any infection can become more invasive, leading to cellulitis, necrotizing fasciitis, scarlet fever, or sepsis. GAS remains sensitive to penicillin antibiotics, which are a mainstay of pharyngitis. However, this monotherapy is typically insufficient during invasive infections, which are highly pro-inflammatory (3) and can require adjunctive antibiotics like clindamycin to limit toxin production, surgical removal of infected tissue, and extensive supportive care

Address correspondence to Christopher N. LaRock, christopher.larock@emory.edu.

The authors declare no conflict of interest.

See the funding table on p. 8.

(4). The case fatality and economic burden of these infections is high in the United States and worse in resource-limited environments where disease is common (5).

One of the major emergent strategies to treat infections is the targeting of virulence factors (6). The secreted cysteine protease Streptococcal pyogenic exotoxin B (SpeB) is a primary virulence factor of GAS and is an attractive drug target for several reasons (7). Among these, SpeB is unique to GAS, but highly conserved within the species, speaking to the possible specificity of its targeting. Furthermore, SpeB is abundant during infection (8–10), and expression correlates with the severity of disease in humans and model infections (11–15). *In vitro* and *in vivo* experiments show that SpeB degrades tissue (16, 17), inactivates immune antimicrobials (15, 18–26), and activates pathological proinflammatory pathways through direct and indirect mechanisms (27–32). These activities cumulatively contribute to both GAS growth and injury to the host (33) and promote pathogenesis in skin (34–36) and upper respiratory (32, 37) infection models in mice. Consequently, inhibiting SpeB may be beneficial during infection.

The aim of this study was to identify small molecule inhibitors of SpeB that would have therapeutic potential for treating infections by GAS. The catalytic cysteine of SpeB, like most other cysteine proteases, is highly reactive and can be inhibited by the most common inhibitors of this family (7, 38). However, broad-spectrum inhibitors would be unsuitable as anti-infectives due to targeting of mammalian proteases leading to toxicity and immunosuppressive effects (39). Furthermore, in addition to looking for strong activity against SpeB, we also considered using known, affordable compounds that could be made available to broad populations. To these ends, we screened a diverse library of off-patent drugs and drug-like compounds that was developed by the Medicines for Malaria Venture (MMV) for the treatment of neglected and poverty-related diseases (40). All compounds in the library have been screened for toxicity (41), a large fraction of them have been tested *in vivo*, and >10% are approved for use in humans for at least one indication. Most potential drugs fail in preclinical or clinical studies; repurposing of existing pharmaceuticals is an avenue to potentially bypass this bottleneck inexpensively, quickly, and safely. We define several lead compounds with therapeutic potential, including pentamidine, a drug already in clinical use for the treatment of protozoal infections.

## MATERIALS AND METHODS

### Bacterial strains

GAS 5448, Δ*speB*, *Staphylococcus aureus* F-182, and their growth are previously described (42–44). Briefly, all bacteria were routinely grown in Todd-Hewitt Yeast (THY; Difco) or on THY-agar at 37°C and 5% $CO_2$. Bacteria from overnight cultures were first washed and diluted in phosphate-buffered saline (PBS) in preparation for infection experiments. Native SpeB was purified from culture supernatants by ammonium sulfate (Sigma) precipitation followed by cleanup by size-exclusion chromatography (ÄKTA FPLC, PE) and Centricon filtration (EMD Millipore), as previously described (26).

### Chemical library

The Pathogen Box compound library was kindly provided by MMV (Switzerland) dissolved in 100% Dimethyl sulfoxide (DMSO) to 1 mM and arrayed in 96-well format and stored at −80°C. Further dilutions were in PBS for screening experiments. Test compounds in this library were selected as having a minimum of fivefold selectivity over mammalian cells for either *Plasmodium*, *Mycobacterium tuberculosis*, or kinetoplastid protists. Twenty-six additional drugs, including conventional antibiotics (e.g., doxycycline, rifampin, streptomycin, and linezolid), anti-infectives against nonbacterial pathogens (e.g., praziquantel, nifurtimox, and miltefosine), and other common drugs (e.g., fluoxetine and auranofin) were included as an additional reference set, distributed throughout the library. The lead hit pentamidine was validated with a compound

purchased from a commercial supplier (Sigma) at 99% purity and diluted in PBS (no DMSO).

## Molecular docking

The interaction of the mature SpeB sequence of PDB:2uzj (45) and the chemical structures of each antibiotic from PubChem was examined using the Chai-1 Model (46). Modeling was agnostic without specified input restraints, and the best-scoring model, all of which were above a ptm of 0.9, was selected for analysis in ChimeraX v1.9. Each molecule was oriented to the same position using Matchmaker for displaying antibiotic occupancy in the substrate cleft of SpeB, and red and blue were used to indicate computed surface electrostatics, as previously described (47). Interacting amino acids of SpeB were identified by the Contacts function of ChimeraX with the default parameter of $>-0.4$ Angstrom van der Waals radii.

## SpeB inhibitor screen

In the initial screen, SpeB activity was measured in the presence of test compound in individual wells of a 96-well plate using the substrate sub103, Mca-IFFDTWK-Dnp (CPC Scientific), essentially as previously described (34). Briefly, test compounds were added to achieve a 100 µM final concentration per well with 10 nM SpeB and 2 mM sub103 (CPC Scientific), all diluted in PBS pH 7.4 with 2 mM dithiothreitol (Sigma) and 0.01% Tween (Sigma). Reactions were incubated at 37°C, and the change of fluorophore excitation at 323 nm and emission at 398 nm after 18 h end-point was measured using a Nivo plate reader (PerkinElmer). A histogram of the frequency distribution was generated using the Prism 10 (GraphPad) Descriptive Statistics function with automated binning. Lead compounds were selected from the lowest activity bin for further verification in a dilution series incubated at 37°C with continuous monitoring of fluorescence under the parameters as above.

## Neutrophil infection assays

Whole human blood was collected from healthy adult donors with informed consent and approval from the Institutional Review Board at Emory University. Blood was collected into heparinized Vacutainer tubes, and primary neutrophils were isolated using Polymorphprep (Axis-Shield). Neutrophils were diluted in Roswell Park Memorial Institute medium (RPMI) containing 10% heat-inactivated fetal bovine serum (FBS) with no antibiotics to reach a final concentration of $10^5$ cells/mL after the addition of $1 \times 10^6$ CFU of GAS (final multiplicity of infection; MOI, of 10) for 60 minutes, with the addition of pentamidine or PBS control, then plated on THY (Todd-Hewitt Media w/ 2% Yeast Extract) agar, essentially as previously (48). CFUs were enumerated after overnight incubation of THY agar plates at 37°C and compared to initial to calculate percentage growth. Values are expressed as means ± standard error unless otherwise specified. Differences were determined by ANOVA with Tukey post-test.

## RESULTS

### Inhibition of SpeB by library compounds

The complex regulation of streptopain/SpeB (49–52) can lead to the identification of compounds whose impact on SpeB is indirect if tested in a culture with live bacteria (53–55). Therefore, we conducted a screen using purified SpeB to focus only on those compounds that directly impact the enzyme's ability to hydrolyze substrates. To monitor the kinetics of proteolysis, we used the internally quenched Förster resonance energy transfer peptide substrate sub103, which is highly selective and only contains a single cleavage site for SpeB (34). Sub103 was incubated with purified SpeB in the presence of 400 compounds, each at 100 µM, and conversion of the chromogenic substrate was measured after 18 hour incubation at 37°C (Fig. 1A). The frequency distribution of values

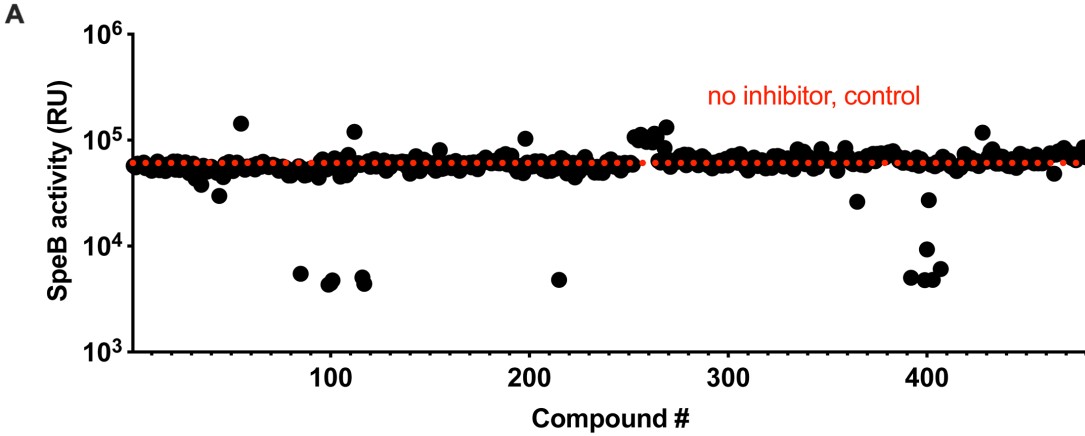

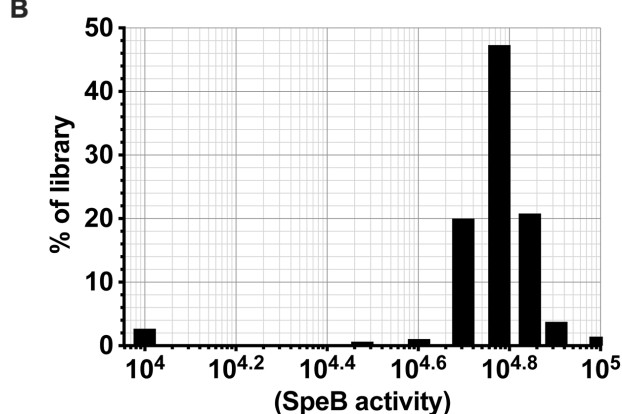

**FIG 1** Graphical representation of results from the primary screen for each of the 400 Pathogen Box compounds. (A) Each individual compound was tested at a concentration of 100 µM for hydrolysis of the substrate sub103, measured in relative fluorescence units (RFU). (B) Distribution of results.

showed nearly all compounds in the library had negligible impact on SpeB activity (Fig. 1B). Ten compounds across two independent runs demonstrated consistent inhibitory activity, for a hit rate of ~2%.

One of these compounds, pentamidine, has known inhibitor activity against proteases of *Porphyromonas gingivalis* (56). Therefore, we first focused on the other compounds for potentially novel activity. Each compound was incubated in dilutions with SpeB, and hydrolysis of sub103 was monitored over 30 minutes. Two compounds (MMV676588 and MMV022478) lacked activity upon further dilution, but the remaining seven (MMV689758, MMV676603, MMV023953, MMV676512, MMV637953, MMV687254, and MMV024101) retained at least partial but statistically significant inhibitory activity to at least 60 µM (Fig. 2). The best compound, MMV689758, retained significant but modest activity to 5 µM, still failing to reach sub-micromolar efficacy, a typical target for antiinfective drugs (39, 41).

To determine whether there was any commonality between SpeB inhibitors, with the expectation that compounds with similar chemical structures could be used to identify additional compounds for testing, we used molecular structure prediction modeling to examine their predicted interaction with SpeB. By unguided analysis, each compound was predicted to occupy the same region in the substrate pocket of SpeB, meaning they could be competitive inhibitors for substrate binding (Fig. 3). There was limited structural homology between compounds. Nonetheless, these data may be useful to inform lead compounds for further chemical refinement, since the ability to occlude the substrate pocket via interaction with its residues would be an expected property of a useful inhibitor.

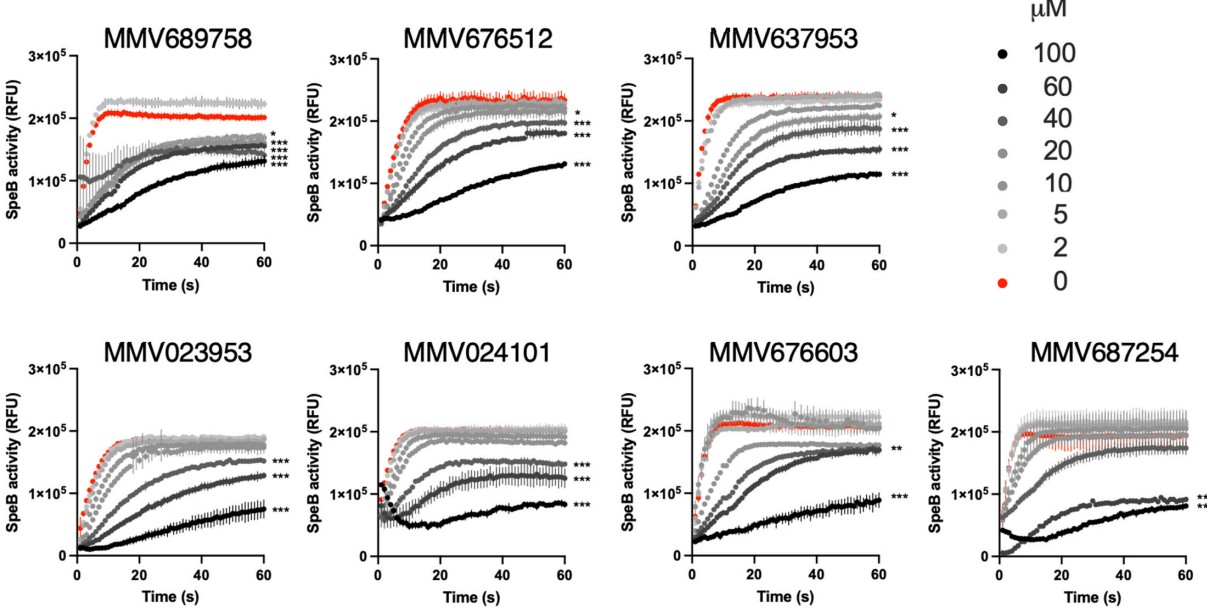

**FIG 2** Activity of candidate compounds against SpeB. Cleavage of sub103 peptide after 30 minute incubation with purified SpeB with each compound at the concentration indicated. $N = 4$, error bars represent standard deviations.

## Inhibition of SpeB by pentamidine

Since this library was targeted for neglected tropical diseases, it additionally contained established drugs including pentamidine (4-[5-(4-carbamimidoylphenoxy)pentoxy]ben-zenecarboximidamide) as controls for any novel compounds contained therewithin. This aromatic diamidine is an effective antimicrobial against several eukaryotic pathogens, but efficacy against SpeB from GAS has not been established. Pentamidine was predicted to occupy the substrate pocket of SpeB similarly to the established inhibitor E64, in its

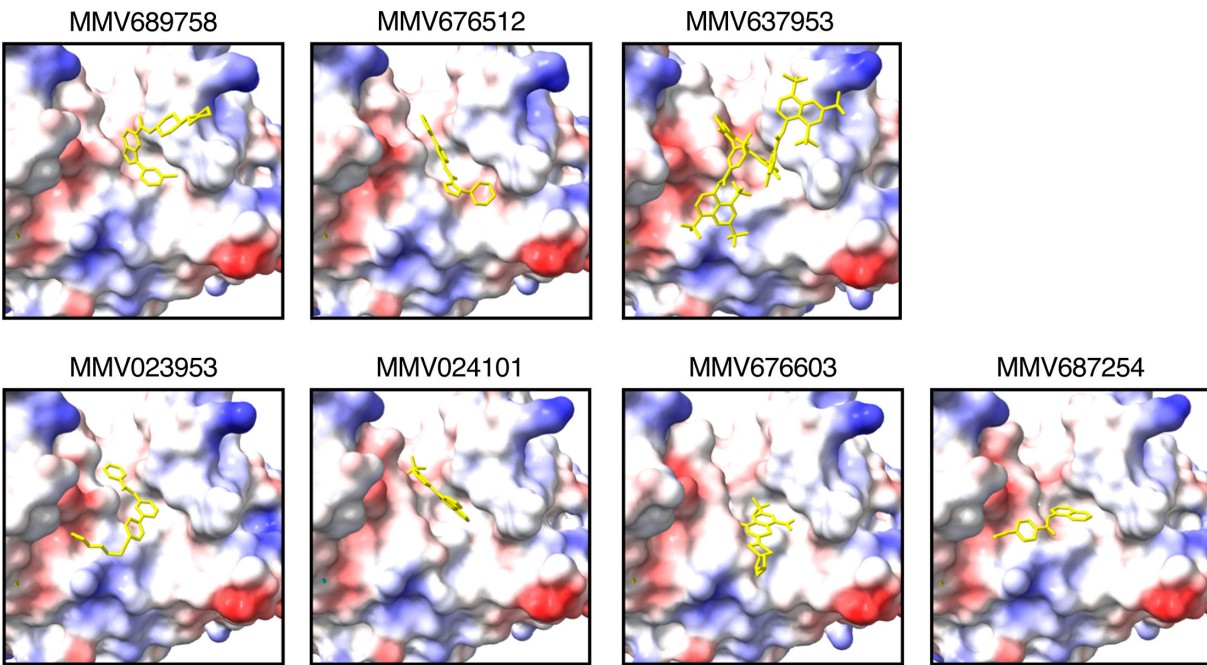

**FIG 3** SpeB-inhibitor interactions. Models of the substrate pocket of SpeB and each indicated compound (yellow). Protein surface electrostatics are colored red and blue for negative and positive charges, respectively, and white color represents neutral residues.

cocrystal structure (45) and by prediction (Fig. 4A through C). Both had possible contacts with the catalytic residue Cys47, but beyond that, interacted differently throughout the binding pocket, with pentamidine further interacting with Asp130, Arg142, Val189, and E64 instead interacting with Ser137, Gln187, and His195. Testing pentamidine's efficacy at inhibiting purified SpeB, we find with titrations of the drug that it has an $IC_{50}$ of 9.671e−009 M *in vitro* (Fig. 4D). This was suggestive that pentamidine could be effective at inhibiting SpeB during an infection.

### *In vitro* activity of pentamidine against *S. pyogenes*

SpeB is important for GAS resistance to innate immune antimicrobials, including those produced by neutrophils (50, 57). To examine whether SpeB inhibition would be useful for sensitizing GAS to neutrophil killing, we modeled this important host-pathogen interaction *in vivo*. 10 µM pentamidine had no impact on the growth of GAS or *S. aureus*, a bacteria included as a control that causes similar infections but lacks a SpeB homolog (Fig. 5A and B). Incubation with neutrophils led to significant decreases in the viability of

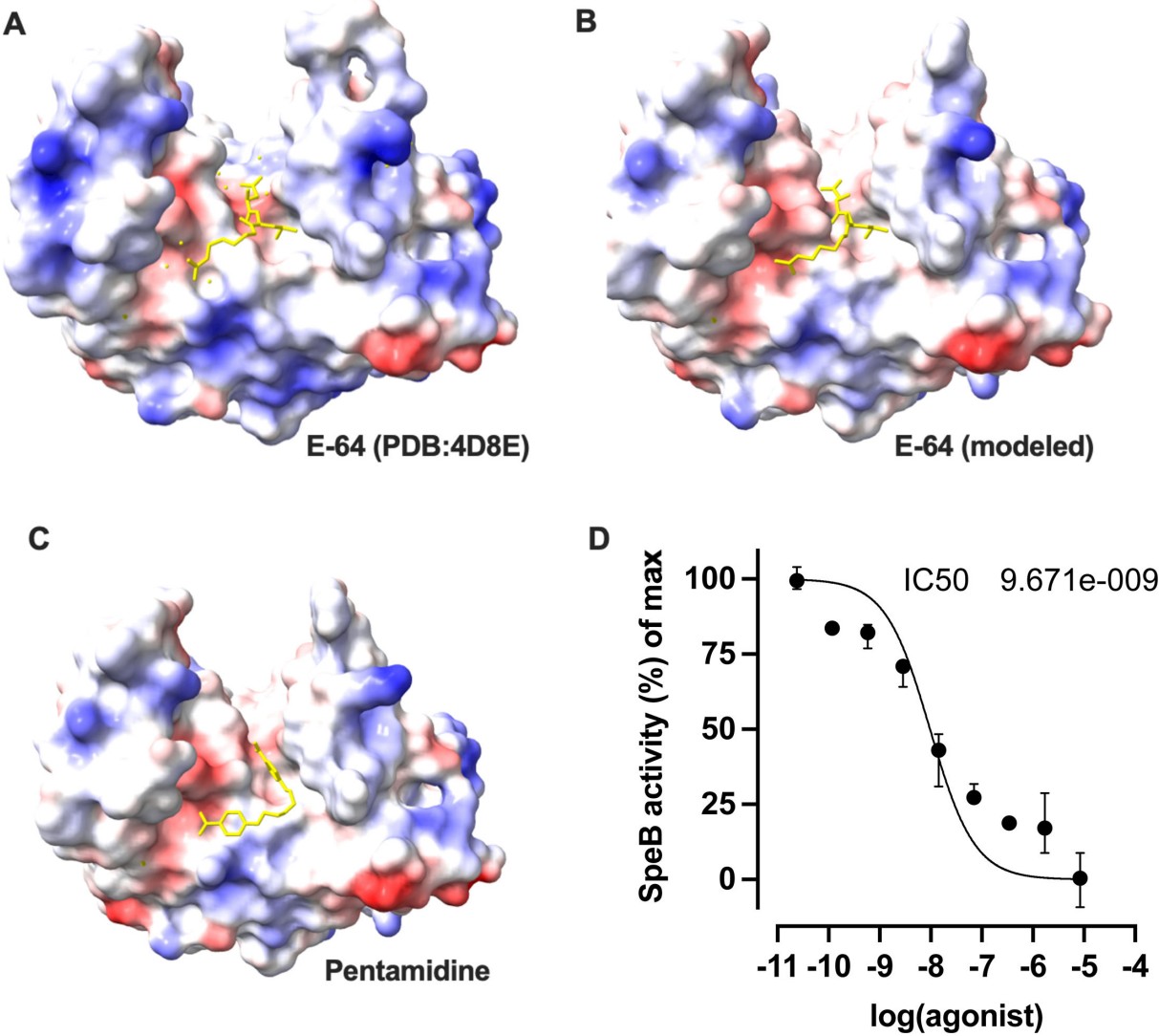

**FIG 4**  Pentamidine inhibits proteolysis by SpeB. (A–C) Models of the substrate pocket of SpeB and each indicated compound (yellow). Protein surface electrostatics are colored red and blue for negative and positive charges, respectively, and white color represents neutral residues. (D) Cleavage of sub103 peptide after 30 minute incubation with purified SpeB was plotted against a range of pentamidine concentrations. $N = 4$, error bars represent standard deviations.

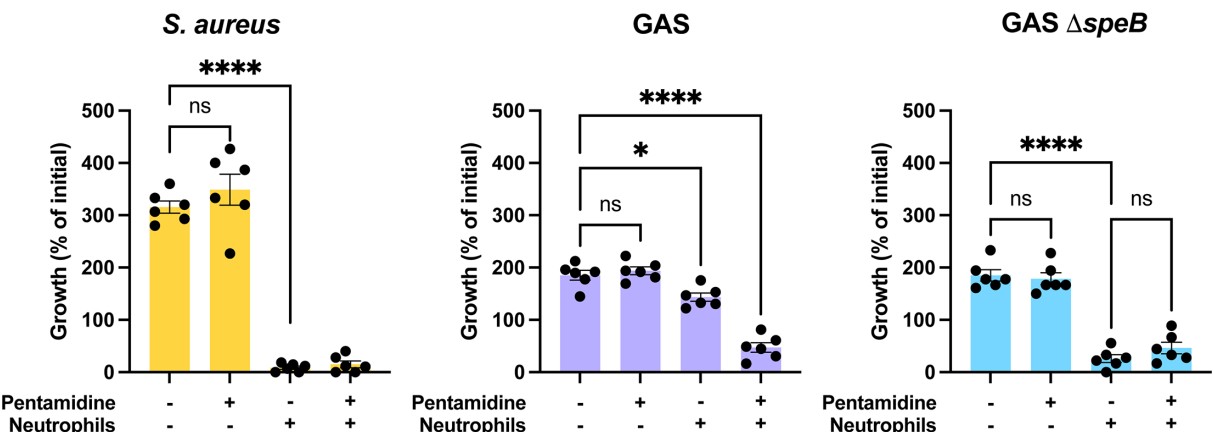

**FIG 5** Pentamidine sensitized GAS to killing by neutrophils. Growth was measured by the change in colony count of GAS or *S. aureus* after 2 hour infection of human primary neutrophils with 10 µM pentamidine treatment; $n = 6$. *$P < 0.05$; ****$P < 0.00005$ (ANOVA; Tukey post-test). Experiments were performed three times; error bars represent standard deviations.

GAS and *S. aureus* (Fig. 5A and B), but only the killing of GAS and not *S. aureus* during incubation with neutrophils was significantly increased by the addition of pentamidine. Additionally, no further killing by neutrophils was observed when Δ*speB* mutant of GAS was treated with pentamidine (Fig. 5C). This suggests a potential for pentamidine to sensitize GAS to killing by the innate immune system.

## DISCUSSION

The Pathogen Box drug library has previously identified new inhibitors against a diverse set of pathogens including *Toxoplasma gondii* (58), *Candida albicans* (59), *G. lamblia, Cryptosporidium parvum* (60), *Vibrio cholerae* (61), *Acinetobacter baumannii* (62), and *Escherichia coli* (63). In our study examining GAS, ~2% of the compounds had significant protease inhibitor activity toward SpeB. A previous screen of 16,000 compounds of the Maybridge HitFinder HTS library had a hit rate of 0.018% and identified a competitive inhibitor (64). Of our top hits, activity against *Plasmodium falciparum* has been shown by MMV689758 and MMV676603 (65, 66) and *M. tuberculosis* by MMV687254 (67). No prior reports showing antimicrobial activities of MMV676512, MMV637953, MMV023953, or MMV024101 could be found. Suramin, MMV637953, has broadly antimicrobial activities with reports establishing efficacy against *Trypanosoma brucei* and other parasitic infections, *C. albicans,* and several viruses (68–70). Consistent with the inhibition of SpeB, among its activities is the inhibition of host proteases, including thrombin, neutrophil proteases, and caspases—activities that would be undesirable in the treatment of GAS infections (68). Altogether, insufficient support could be found for a common structure between inhibitor compounds.

Pentamidine is a cationic aromatic diamine, which is known to interfere with polyamine synthesis and RNA polymerase activity in protozoal cells. It is typically delivered by oral, intravenous, and intramuscular routes, by topical delivery for skin disease and aerosolized delivery for pulmonary diseases with some success (71, 72). These results argue for another potential utility for its use. Consistent with our findings, pentamidine has been observed to be an inhibitor of the gingipains of *P. gingivalis* (56). These secreted virulence factors are, like streptopain, abundant and highly active cysteine proteases that target multiple host proteins to allow the bacterium to resist killing by neutrophils and other immune effector mechanisms (73). Diverse pathogens other than *P. gingivalis* and GAS encode cysteine protease virulence factors, leaving the possibility that pentamidine may be more broadly useful as an antimicrobial.

While other inhibitors of SpeB exist, such as the epoxide E-64 (N-[N-(L-3-trans-carbox-yirane-2-carbonyl)-L-leucyl]-agmatine), their broad-spectrum activities can also inhibit

essential host proteases (34). That pentamidine is already in clinical use argues that it can be tolerated, though with possible severe adverse events reported that include hypotension and nephrotoxicity. It has additionally been seen to prevent IL-1 cleavage, suggesting it can inhibit the host cysteine protease caspase-1 (74), without other obvious effects on the inflammasome (75). However, if this does occur in physiologically relevant conditions, it is likely to still not impair the immune responses because GAS is known to activate IL-1β and related cytokines independent of caspase-1 and other inflammasome-associated proteases (28, 34), and in some infection modes may even be protective (32). Additional experiments would be required to determine the breadth of pentamidine's protease inhibitor activities, modality of inhibition, and whether resistance can evolve. Lastly, there is a significant overlap in the burden of the tropical diseases for which pentamidine is used and the burden of GAS. The long duration of pentamidine administration to similar concentrations as typically required for therapeutic or prophylactic purposes for these diseases (76) may thus have the possibility to impact whether an individual is co-infected by GAS. No existing clinical data could be found to support a correlation, which can be addressed in further studies examining disease incidence in these populations.

## ACKNOWLEDGMENTS

We thank the Medicines for Malaria Venture foundation (MMV; Switzerland) for providing the Pathogen Box compounds that allowed this study. We appreciate the technical support provided by Children's Healthcare of Atlanta and Emory University's Children's Clinical and Translational Discovery Core for whole blood and cell processing, and the human donors. Molecular graphics and analyses performed with UCSF ChimeraX, developed by the Resource for Biocomputing, Visualization, and Informatics at the University of California, San Francisco, with support from NIH R01-GM129325 and the Office of Cyber Infrastructure and Computational Biology, NIAID. C.N.L. is supported by National Institutes of Health grants R01-AI153071 and R01-AI180089. C.N.L. is a Burroughs Wellcome Fund Investigator in the Pathogenesis of Infectious Disease. The content of this publication is solely the responsibility of the authors and does not necessarily represent the official views of any of its funders. No funders contributed to the study design or conclusions.

K.T. and C.L. conducted the studies, wrote the manuscript, and approved the final manuscript.

## AUTHOR AFFILIATIONS

[1]Department of Biology, Emory University, Atlanta, Georgia, USA

[2]Department of Microbiology and Immunology, Emory University School of Medicine, Atlanta, Georgia, USA

[3]Department of Medicine, Division of Infectious Disease, Emory University School of Medicine, Atlanta, Georgia, USA

[4]Antimicrobial Resistance Center, Emory University, Atlanta, Georgia, USA

## AUTHOR ORCIDs

Christopher N. LaRock  http://orcid.org/0000-0003-3035-5331

## FUNDING

| Funder | Grant(s) | Author(s) |
| --- | --- | --- |
| National Institutes of Health | AI153071, AI180089 | Christopher N. LaRock |
| Burroughs Wellcome Fund | Investigator in the Pathogenesis of Infectious Disease | Christopher N. LaRock |

## AUTHOR CONTRIBUTIONS

Keya Trivedi, Conceptualization, Data curation, Formal analysis, Investigation, Methodology, Validation, Visualization, Writing – original draft, Writing – review and editing | Christopher N. LaRock, Conceptualization, Data curation, Formal analysis, Funding acquisition, Investigation, Methodology, Project administration, Supervision, Validation, Visualization, Writing – original draft, Writing – review and editing

## ADDITIONAL FILES

The following material is available online.

### Open Peer Review

**PEER REVIEW HISTORY (review-history.pdf).** An accounting of the reviewer comments and feedback.

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
