## [Reviewer comments · Microbiology Spectrum]

Microbiology Spectrum

Pentamidine inhibition of streptopain attenuates *Streptococcus pyogenes* virulence

Keya Trivedi and Christopher LaRock

Corresponding Author(s): Christopher LaRock, Emory University School of Medicine

Review Timeline:

Submission Date:	March 12, 2025
Editorial Decision:	April 14, 2025
Revision Received:	April 23, 2025
Accepted:	June 2, 2025

Editor: Justin Kaspar

Reviewer(s): The reviewers have opted to remain anonymous.

Transaction Report:

DOI: <https://doi.org/10.1128/spectrum.00758-25>

Re: Spectrum00758-25 (**Pentamidine inhibition of streptopain attenuates *Streptococcus pyogenes* virulence**)

Dear Dr. Christopher N. LaRock:

Thank you for the privilege of reviewing your work. Below you will find my comments, instructions from the Spectrum editorial office, and the reviewer comments.

The referenced manuscript has now been evaluated by two reviewers. Both reviewers are supportive of the work, and have provided several comments to improve the manuscript prior to publication. As you will see, most of the comments provided relate to modification of the text / improvement of the figures shown or further clarification of the figure legends. Please note that Reviewer 2's comments are provided in an attachment that should be attached to this email.

Revision Guidelines

Sincerely,
Justin Kaspar
Editor
Microbiology Spectrum

Reviewer #1 (Comments for the Author):

line 93: "Pathogen Box library compound" should read "Pathogen Box compound library"

line 94: Please indicate which DMSO concentration was used to dissolve the compounds.

line 101: "high purity compounded" should read "high purity compound"

I find Figure 1B more confusing than helpful. How was the binning performed? What is meant by SpeB activity x 1000? Please clarify.

In general, the description of the experiments and results in each individual paragraph of the results section seems too short to me. Please go into more detail about the results in the text.

Figure 2/lines 150-157: Do the headings of the diagrams refer to the compounds in the Pathogen Box? Why was a different scaling of the y-axis used for the compound MMV023953 than for the other compounds? At which concentrations was the inhibitory effect of the tested compounds statistically significant? The standard deviations are very large, especially for MMV022478.

Figure 3/lines 159-165: Please elaborate on the structural features of the compounds and how they interact with the SpeB substrate binding pocket. How do you deduce from figure 2 that the respective compounds are competitive inhibitors for substrate binding? Are there structural similarities to pentamidine?

Figure 4/lines 167-174: Obviously there are structural similarities between pentamidine and E64. Please expand on this. What is the unit of the indicated IC50?

Figure 5/lines 176-184: Please describe the experiment/results properly! In line 179 the 10 μ M refer to pentamidine, I assume? Please indicate.

Discussion, line 196-197: In the results section the authors propose competitive inhibition as the mode of action for all compounds. Here they state that there is no commonality in mechanism between the compounds. Please explain.

line 199-200: This sentence is incomprehensible to me.

Pentamidine has a number of severe side effects. This should be mentioned in the discussion. How would the authors proceed in potentially establishing pentamidine as an anti GAS drug or is this even wanted? What about the other potential inhibitors? Are they worth pursuing? How could they be developed into SpeB targeting drugs?

Reviewer #2 (Comments for the Author):

From document summary "The study has strong screening and initial characterization studies. It is not clear if pentamidine was part of the library or was used later on. It may benefit from revising later assays such as those found in figures 4 and 5 to strengthen the inhibitor capabilities and specificity while removing concerns about toxicity. The discussion can be improved by focusing on pentamidine, SpeB, and *S. pyogenes* as this would bring the manuscript full circle."

Summary

The manuscript titled “**Pentamidine inhibition of streptopain attenuates *Streptococcus pyogenes* virulence**” provides a description of the process to identify a molecule, pentamidine as a potential inhibitor of the cysteine protease SpeB in the human pathogen *S. pyogenes*. SpeB is a well-known and important virulence factor associated with *S. pyogenes* pathogenesis. The manuscript provides as part of the hypothesis that targeting SpeB can provide an advantage to the immune system.

The manuscript provides information about a molecule library and tests used to narrow and isolate compounds based on SpeB activity inhibition. It also has follow up tests to determine potential efficacy in protein-molecule and host cell studies. The study has strong screening and initial characterization studies. It is not clear if pentamidine was part of the library or was used later on. It may benefit from revising later assays such as those found in figures 4 and 5 to strengthen the inhibitor capabilities and specificity while removing concerns about toxicity. The discussion can be improved by focusing on pentamidine, SpeB, and *S. pyogenes* as this would bring the manuscript full circle. Please find below comments using the manuscript’s line numbers.

Abstract

Line 23: For clarity, change to *Streptococcus pyogenes* (also known as GAS; Group A *Streptococcus*)

Importance

Line 40: similar to line 23

Line 46: At this point in the document, the abstract did not specify that GAS was unable to gain resistance to pentamidine. Would suggest keeping the focus on the positives of not altering the microbiota (specifically bacteria) too much.

Line 48: use GAS for consistency

Introduction

Line 56: move “strep throat” to line 52. Mentioned there first.

Line 57: provide 1 example of an adjunctive antibiotic.

Line 59: the fatality is mentioned as well in the abstract, more specifically towards resource-limited. It may be helpful to place a reference and statistics regarding this to support the section.

Line 59: a transition may assist with the introduction of SpeB in the next section.

Line 61: For clarity, it may be important to note that SpeB is regulated. There are also clinical strains with high SpeB production due to mutations. Perhaps adding these references may strengthen this section and provide more context to why this protein was selected.

Line 76: perhaps redefine the sentence to 'using known, affordable?, and readily available antimicrobials'. I'm not sure what the statement 'populations where it would be most needed' means.

Line 78: Reference needed for the Medicines for Malaria Venture

Line 82: Should the end sentence be something about pentamidine to close and bring back full circle?

Materials and Methods:

Line 87: Add broth after Todd-Hewitt Yeast. Perhaps indicate if both strains used the exact same culturing method.

Line 93: For libraries provide the % DMSO as it can affect bacterial growth at higher concentrations. Would indicate the purity level for your prepared library.

Line 95: Provide the dilution used, unless it is stated later in a different method.

Line 96: Provide equation for selectivity if used.

Line 97: Provide if the reference set followed the same format as the library setup.

Line 101: Given that the target molecule is mentioned here, it may be important to have a section dedicated to the validation process and final purity of pentamidine.

Line 106: Please specify what is the cutoff for the 'best scoring model'.

Line 111-116: Please specify if the test was end-point or kinetic. The current information (without reading the results or looking at figures) does not explain the protocol. Is this a micro titer read, an incubator setup, timeframe of assay, etc. When it says 'comparable to control' is there a cut off, a percent, etc? Could be clearer to the reader. Your lines 118-123 are more in line with what was suggested here.

Line 123: Cutoff, number of times done, and any equations regarding the outputs would make this section clearer.

Line 126: Is the IRB approval something required as supplemental for *ASM Spectrum*?

Line 129: CFU/ML. Number of times done, duplicates or triplicates, and equations for evaluation would make the section more complete.

Line 133: is this section for everything above?

Results (Figure notes added as they appear in the Results text for context)

Line 144: Based on the figure, there is a range for the 9 compounds, it would be useful to provide the range as context for why pentamidine was selected as the final. Given that the manuscript is specific to pentamidine, its introduction at this point would be ideal so that the story of its characterization can unfold in the later sections.

Figure 1A: is Pathogen compounds the same as the Medicines for Malaria Venture? Or did you mean Pathogen Box as written in Methods? If yes, would replace for consistency. What is (RU), not defined in the legend. 1A is the average of the duplicate test, correct?

Figure 1B: X and Y axis can be further clarified. Not sure why the 20 in the X-axis has a plus sign and the others do not.

Line 147: *from this library*. This reason has a slightly different weight. Would saying bioactivity and accessibility keep consistency? Not sure if the sentence is needed in this section.

Line 154: The compounds have a variety of outcomes. It may be helpful to provide clusters. For example, there are some that seem very concentration-dependent while others have a cut-off. How do all compare to pentamidine is also important to note. This would raise the question later as to which type of graph is most valuable for moving to host conditions.

Line 155: Provide average micromolar range and/or reference.

Figure 2: Define the Y axis in the legend. The methods do not indicate a 30 min timeframe nor does the graph, clarification needed. The image could be larger to allow for better view of each concentration line with standard deviations. Some of the graphs are not aligned properly. One of the graphs, MMV023953 has a different Y-axis. Perhaps using scientific notation may help clean the Y-axis a bit. The graph for pentamidine could be boxed or highlighted here. Should a known inhibitor be used here as a control?

Line 161: *we used*

Line 163: Define the region for binding, include reference. Do some seem to be more closely associated to positive charges in the region?

Figure 3: The modeling for pentamidine could be boxed or highlighted here. Could you check if image MMV637953 and MMV023953 are not the same? The image is small but they look quite similar.

Figures 2/3: Based on figure 2, there are some that were better overall, would it be useful for the audience to have the compounds arranged from best to worst in these 2 figures?

Line 167: the term “neglected tropical diseases” is used for the first time. As mentioned earlier for consistency. It is mentioned here that pentamidine was a positive control to the library. I am not sure this was indicated earlier. Up to this moment in the document, I had assumed pentamidine was one of the 400 compounds. Would you clarify the text please?

Line 169: Since it has not been tested in GAS, it may be important to note any culture toxicity using the same concentration gradient used for the earlier test (as observed in figure 2). This would also help in indicating if pentamidine can cross the cell membrane.

Line 169: Would it strengthen the information provided to show that SpeB concentrations (only activity) is not affected in cultures using Western Blots?

Line 173: Should it indicate that it is effective *in vitro*? Could it be referenced if the concentration found here is ideal in comparison to host cells?

Figure 4A-C: These would go well in the previous image, and it includes the known inhibitor E64.

Figure 4D: Not able to observe any error bars, is sentence part of this graph? I think there's an additional 0 in e-009. Text in results says e-09.

Line 179: It is mentioned here that 10 μM does not affect growth before addition of neutrophils. This further supports the recommendation above for lines 169 and 173. Is the 10 μM the lowest concentration that can be used?

Line 180: When the manuscript says “species-specific”, does it refer to *S. aureus* also having secreted proteases? Otherwise, based on the information provided, it is not clear why *S. aureus* is a positive control if it's specific to SpeB. I guess the question would be, what is pentamidine binding to in the *S. aureus* assay? Would it be more relatable to have other Streptococcal species for comparison, ones perhaps with close or distantly related proteases?

The assay could benefit from using a deletion of SpeB to determine if the decrease is due specifically to the interaction with SpeB and not general toxicity.

Figure 5: Have Figure legend match the Y-axis. Does each black dot represent the experiments for a total of 6? Not sure error bars are present.

Discussion

Line 193: Perhaps arrange by category – parasites, fungal, bacterial, etc.

Line 197-199: There's no transition here to connect with pentamidine.

Line 199: I think there is a word missing '*which enters is known*'...

Line 202: should say '*with some success*'

Line 208: Perhaps indicate that because there might be a broader target (not just *S. pyogenes*) a study of related SpeB or SpeB-like proteins in commensal organisms may be useful given the desire to minimally target the biome?

Line 212-218: I think here the concentrations are the point. Please refer to the above comments on host cell concentrations please.

Other comments: Some future questions that perhaps could be part of the discussion include is pentamidine binding reversible / irreversible, is there known information about resistance generated in other microorganisms that could be a concern for *S. pyogenes*, would they observe similar results if the assay was done in mice (% survival), etc.

Acknowledgements

Line 227: Any information regarding the IRB?

References

Some have the DOI and some do not. Some have the Genus and species italicized and some do not. Quick check recommended.

Reviewer #1 (Comments for the Author):

We thank the reviewer for their helpful comments, each is addressed as requested line-by-line in the revision and detailed below.

line 93: "Pathogen Box library compound" should read "Pathogen Box compound library"

Corrected as requested

line 94: Please indicate which DMSO concentration was used to dissolve the compounds.

Corrected as requested

line 101: "high purity compounded" should read "high purity compound"

Corrected as requested

I find Figure 1B more confusing than helpful. How was the binning performed? What is meant by SpeB activity x 1000? Please clarify.

This figure is used to illustrate the cutoff for which compounds were identified as inhibitors. We have simplified this figure so that the activity (x-axis) matches Fig 1A, and the y-axis has been changed to % of compounds to match the hit rate (2%) mentioned in the text. Details have been added to the methods to read "A histogram of the frequency distribution was generated using the Prism 10 (Graphpad) Descriptive Statistics function with automated binning. Lead compounds were selected from the lowest activity bin for further verification" (line 117) and text clarified to read "The frequency distribution of values showed nearly all compounds in the library had negligible impact on SpeB activity (Figure 1B). 10 compounds across 2 independent runs demonstrated consistent inhibitory activity, for a hit rate of ~2%."

In general, the description of the experiments and results in each individual paragraph of the results section seems too short to me. Please go into more detail about the results in the text.

The requested details have been expanded upon broadly as requested

Figure 2/lines 150-157: Do the headings of the diagrams refer to the compounds in the Pathogen Box? Why was a different scaling of the y-axis used for the compound MMV023953 than for the other compounds? At which concentrations was the inhibitory effect of the tested compounds statistically significant? The standard deviations are very large, especially for MMV022478.

There was no reason for the different scaling, this has been corrected as requested. The large standard deviations for MMV022478 are due to the poor solubility and aggregative properties of this compound, a pyrazolopyrimidine. These characteristics make it a poor lead compound, other than just being a poor inhibitor of SpeB, justifying us excluding it from further study.

Figure 3/lines 159-165: Please elaborate on the structural features of the compounds and how they interact with the SpeB substrate binding pocket. How do you deduce from figure 2 that the respective compounds are competitive inhibitors for substrate binding? Are there structural similarities to pentamidine?

We agree with the reviewer and have elaborated more on this, this was the motivation to followup on pentamidine and separate it from the other compounds. Regarding competitive inhibition, we have reordered these sentences to make this more clear "By unguided analysis, each compound was predicted to occupy the same region in the substrate pocket of SpeB, meaning they could be competitive inhibitors for substrate binding (Figure 3). There was limited structural homology between compounds, but any occlusion of the substrate pocket via interaction with its residues has the possibility to interfere cleavage as seen in enzyme assays (Figure 2)."

Figure 4/lines 167-174: Obviously there are structural similarities between pentanmidine and E64. Please expand on this. What is the unit of the indicated IC50?

We requested, we have expanded on this "Both had possible contacts with the catalytic residue Cys47, but beyond that interacted differently throughout the binding pocket, with pentamidine further interacting with Asp130, Arg142, Val189, and E64 instead interacting with Ser137, Gln187, and His195." with the methods "Interacting amino acids of SpeB were identified by the Contacts function of ChimeraX with the default >-0.4 Å VDW parameter." The unit has

been add, "M".

Figure 5/lines 176-184: Please describe the experiment/results properly! In line 179 the 10 μ M refer to pentamidine, I assume? Please indicate.

The reviewer is correct. "pentamidine" had been added as required, and the statement "during incubation with neutrophils" added to clarify the difference in the final experimental comparison

Discussion, line 196-197: In the results section the authors propose competitive inhibition as the mode of action for all compounds. Here they state that there is no commonality in mechanism between the compounds. Please explain.
line 199-200: This sentence is incomprehensible to me.

This has been clarified as requested. A previous screen had identified a competitive inhibitor. When we refer to no commonality, we intended to refer only to compound structure, and have changed this accordingly "Altogether insufficient support could be found for a common structure between inhibitor compounds."

Pentamidine has a numerous severe side effects. This should be mentioned in the discussion. How would the authors proceed in potentially establishing pentamidine as an anti GAS drug or is this even wanted? What about the other potential inhibitors? Are they worth pursuing? How could they be developed into SpeB targeting drugs?

As requested, we have added the line "That pentamidine is already in clinical use argues that it can be tolerated, though with possible severe adverse events reported that include hypotension and nephrotoxicity". In the final paragraph we discuss the caveats of SpeB inhibitors and focus that since pentamidine is in clinical use already, this could impact Strep co-infections of these patients. Since this is a basic, in vitro study, the data are insufficient to support this as an anti-GAS drug so we do not wish to claim it as such.

Reviewer #2 (Comments for the Author):

Summary

The manuscript titled "Pentamidine inhibition of streptolysin A attenuates Streptococcus pyogenes virulence" provides a description of the process to identify a molecule, pentamidine as a potential inhibitor of the cysteine protease SpeB in the human pathogen S. pyogenes. SpeB is a well-known and important virulence factor associated with S. pyogenes pathogenesis. The manuscript provides as part of the hypothesis that targeting SpeB can provide an advantage to the immune system.

The manuscript provides information about a molecule library and tests used to narrow and isolate compounds based on SpeB activity inhibition. It also has follow up tests to determine potential efficacy in protein-molecule and host cell studies. The study has strong screening and initial characterization studies. It is not clear if pentamidine was part of the library or was used later on. It may benefit from revising later assays such as those found in figures 4 and 5 to strengthen the inhibitor capabilities and specificity while removing concerns about toxicity. The discussion can be improved by focusing on pentamidine, SpeB, and S. pyogenes as this would bring the manuscript full circle. Please find below comments using the manuscript's line numbers.

We thank the reviewer for their helpful comments, each is addressed as requested line-by-line in the revision and detailed below.

Abstract

Line 23: For clarity, change to Streptococcus pyogenes (also known as GAS; Group A Streptococcus)

Corrected as requested

Importance

Line 40: similar to line 23

Corrected as requested

Line 46: At this point in the document, the abstract did not specify that GAS was unable to gain resistance to pentamidine. Would suggest keeping the focus on the positives of not altering the microbiota (specifically bacteria) too much.

We agree with the reviewer and correct this as requested

Line 48: use GAS for consistency

Corrected as requested

Introduction

Line 56: move "strep throat" to line 52. Mentioned there first.

Corrected as requested

Line 57: provide 1 example of an adjunctive antibiotic.

Corrected as requested "like clindamycin to limit toxin production"

Line 59: the fatality is mentioned as well in the abstract, more specifically towards resource-limited. It may be helpful to place a reference and statistics regarding this to support the section.

Corrected as requested, with reference "The case fatality and economic burden of these infections is high in the United States and worse in resource-limited environments where disease is common (5)."

Line 59: a transition may assist with the introduction of SpeB in the next section.

Transition add as recommended, with citation "One of the major emergent strategies to treat infections is the targeting of virulence factors (6)."

Line 61: For clarity, it may be important to note that SpeB is regulated. There are also clinical strains with high SpeB production due to mutations. Perhaps adding these references may strengthen this section and provide more context to why this protein was selected.

This paragraph has been revised accordingly, including "expression correlates with the severity of disease", "but highly conserved within the species, speaking to the possible specificity of its targeting", and a closing statement "Consequently, inhibiting SpeB may be beneficial during infection."

Line 76: perhaps redefine the sentence to 'using known, affordable?, and readily available antimicrobials' . I'm not sure what the statement 'populations where it would be most needed' means.

Corrected as requested "using known, affordable compounds that could be made available to broad populations."

Line 78: Reference needed for the Medicines for Malaria Venture

Corrected as requested

Line 82: Should the end sentence be something about pentamidine to close and bring back full circle?

Corrected as requested, " , including pentamidine, a drug already in clinical use for the treatment of protozoal infections."

Materials and Methods:

Line 87: Add broth after Todd-Hewitt Yeast. Perhaps indicate if both strains used the exact same culturing method.

Corrected as requested, "all"

Line 93: For libraries provide the % DMSO as it can affect bacterial growth at higher concentrations. Would indicate the purity level for your prepared library.

Corrected as requested "100%". We note that the library was not examined for bacterial growth, and pentamidine experiments contained PBS as a buffer, not DMSO (methods)

Line 95: Provide the dilution used, unless it is stated later in a different method.

Added as requested, "dissolved in 100% DMSO to 1 mM"

Line 96: Provide equation for selectivity if used.

Details on the selection of compounds for additional validation have been added "A histogram of the frequency distribution was generated using the Prism 10 (Graphpad) Descriptive Statistics function with automated binning. Lead compounds were selected from the lowest activity bin for further verification"

Line 97: Provide if the reference set followed the same format as the library setup.

Added as requested, "distributed throughout the library."

Line 101: Given that the target molecule is mentioned here, it may be important to have a section dedicated to the validation process and final purity of pentamidine.

Added as requested, "at 99% purity and diluted in PBS (no DMSO)."

Line 106: Please specify what is the cutoff for the 'best scoring model'

Added as requested, "all were above a ptm of 0.9"

Line 111-116: Please specify if the test was end-point or kinetic. The current information (without reading the results or looking at figures) does not explain the protocol. Is this a micro titer read, an incubator setup, timeframe of assay, etc. When it says 'comparable to control' is there a cut off, a percent, etc? Could be clearer to the reader. Your lines 118-123 are more in line with what was suggested here.

This section was been rewritten with the additions requested, "In the initial screen, SpeB activity was measured in the presence of test compound in individual wells of a 96 well plate using the substrate sub103, Mca-IFFDTWK-Dnp (CPC Scientific), essentially as previously described (34). Briefly, test compounds were added to achieve a 100 μ M final concentration per well with 10 nM SpeB and 2 mM sub103 (CPC Scientific), all diluted in PBS pH 7.4 with 2 mM dithiothreitol (Sigma) and 0.01% Tween (Sigma). Reactions were incubated at 37°C and the change of fluorophore excitation at 323 nm and emission at 398 nm after 18 h end-point was measured using a Nivo plate reader (PerkinElmer). A histogram of the frequency distribution was generated using the Prism 10 (Graphpad) Descriptive Statistics function with automated binning. Lead compounds were selected from the lowest activity bin for further verification in dilution series incubated at 37°C with continuous monitoring of fluorescence under the parameters as above."

Line 123: Cutoff, number of times done, and any equations regarding the outputs would make this section clearer.

Details have been added to the methods to read "A histogram of the frequency distribution was generated using the Prism 10 (Graphpad) Descriptive Statistics function with automated binning. Lead compounds were selected from the lowest activity bin for further verification" and text clarified to read "The frequency distribution of values showed nearly all compounds in the library had negligible impact on SpeB activity (Figure 1B). 10 compounds across 2 independent runs demonstrated consistent inhibitory activity, for a hit rate of ~2%."

Line 126: Is the IRB approval something required as supplemental for ASM Spectrum?

The IRB approval letter will be made available to journal staff if asked; I have never encountered this at any journal. Use of blood from anonymous donors with no genetic or other identifying component is considered human subject exempt research by US and ASM standards.

Line 129: CFU/ML. Number of times done, duplicates or triplicates, and equations for evaluation would make the section more complete.

We have changed the grammar here to make this clear “Neutrophils were diluted in RPMI containing 10% FBS with no antibiotic to reach a final concentration of 105 cells/mL after the addition of 1×10^6 CFU of GAS (final MOI = 10) for 60 minutes, with the addition of pentamidine or PBS control,” and clarified “and compared to initial to calculate percentage growth”. Details on n and repetition are in the corresponding legend for Fig 5.

Line 133: is this section for everything above?

The statistics have been updated to remove tests not performed

Results (Figure notes added as they appear in the Results text for context)

Line 144: Based on the figure, there is a range for the 9 compounds, it would be useful to provide the range as context for why pentamidine was selected as the final. Given that the manuscript is specific to pentamidine, its introduction at this point would be ideal so that the story of its characterization can unfold in the later sections.

Added as requested, a greater discussion as been added where “One of these compounds, pentamidine, has known inhibitor activity against proteases of Porphyromonas gingivalis (72). Therefore, we first focused on the other compounds for potentially novel activity.”

Figure 1A: is Pathogen compounds the same as the Medicines for Malaria Venture? Or did you mean Pathogen Box as written in Methods? If yes, would replace for consistency.

Corrected as requested, “Pathogen Box”

What is (RU), not defined in the legend. 1A is the average of the duplicate test, correct?

Corrected as requested, changed to “RFU” for consistency with other figures, and definition added to legend (line 603), and added details on analysis added to methods

Figure 1B: X and Y axis can be further clarified. Not sure why the 20 in the X-axis has a plus sign and the others do not.

This figure is used to illustrate the cutoff for which compounds were identified as inhibitors. We have simplified this figure so that the activity (x-axis) matches Fig 1A, and the y-axis has been changed to % of compounds to match the hit rate (2%) mentioned in the text. Details have been added to the methods to read “A histogram of the frequency distribution was generated using the Prism 10 (Graphpad) Descriptive Statistics function with automated binning. Lead compounds were selected from the lowest activity bin for further verification” (line 117) and text clarified to read “The frequency distribution of values showed nearly all compounds in the library had negligible impact on SpeB activity (Figure 1B). 10 compounds across 2 independent runs demonstrated consistent inhibitory activity, for a hit rate of ~2%.”

Line 147: from this library. This reason has a slightly different weight. Would saying bioactivity and accessibility keep consistency? Not sure if the sentence is needed in this section.

Agreed, this sentence has been removed for flow and consistency as recommended

Line 154: The compounds have a variety of outcomes. It may be helpful to provide clusters. For example, there are some that seem very concentration-dependent while others have a cut-off. How do all compare to pentamidine is also important to note. This would raise the question later as to which type of graph is most valuable for moving to host conditions.

On analysis of these compounds, there was an insufficient number to power for clustering analysis, and this could introduce a bias against compound classes. As also suggested by reviewer 1, we have re-worked the explanation of the library screen and the transition to the followup studies with pentamidine

Line 155: Provide average micromolar range and/or reference.

Added as requested, “partial inhibitory activity up to at least 20 uM”

Figure 2: Define the Y axis in the legend. The methods do not indicate a 30 min timeframe

nor does the graph, clarification needed. The image could be larger to allow for better view of each concentration line with standard deviations. Some of the graphs are not aligned properly. One of the graphs, MMV023953 has a different Y-axis. Perhaps using scientific notation may help clean the Y-axis a bit. The graph for pentamidine could be boxed or highlighted here. Should a known inhibitor be used here as a control?

Added as requested, we have amended the Y-axis (to match Fig 1), enlarged the images, aligned the graphs, used scientific notation on the Y-axis, and added experimental details and further discussion of these compounds to the text "One of these compounds, pentamidine, has known inhibitor activity against proteases of Porphyromonas gingivalis (72). Therefore, we first focused on the other compounds for potentially novel activity. Each compound was incubated in dilutions with SpeB and hydrolysis of sub103 monitored over 30 minutes. Two compounds (MMV676588 and MMV022478) lacked activity upon further dilution, but the remaining 7 (MMV689758, MMV676603, MMV023953, MMV676512, MMV637953, MMV687254, MMV024101) retained at least partial but statistically significant inhibitory activity to at least 60 μ M (Figure 2). The best compound, MMV689758, retained significant but modest activity to 5 μ M, still failing to reach sub-micromolar efficacy, a typical target for anti-infective drugs (39, 41)."

Line 161: we used

Corrected as requested, "we" (line 271)

Line 163: Define the region for binding, include reference. Do some seem to be more closely associated to positive charges in the region?

Added as requested, we "Both had possible contacts with the catalytic residue Cys47, but beyond that interacted differently throughout the binding pocket, with pentamidine further interacting with Asp130, Arg142, Val189, and E64 instead interacting with Ser137, Gln187, and His195."

Figure 3: The modeling for pentamidine could be boxed or highlighted here. Could you check if image MMV637953 and MMV023953 are not the same? The image is small but they look quite similar.

The reviewer is correct, we appreciate the catch. Due to their similar names "953" there was an error in compiling this figure. This image for MMV637953 has been corrected, and all other data re-examined for accuracy.

Figures 2/3: Based on figure 2, there are some that were better overall, would it be useful for the audience to have the compounds arranged from best to worst in these 2 figures?

As requested, we have arranged these 2 figures from "best to worst"

Line 167: the term "neglected tropical diseases" is used for the first time. As mentioned earlier for consistency. It is mentioned here that pentamidine was a positive control to the library. I am not sure this was indicated earlier. Up to this moment in the document, I had assumed pentamidine was one of the 400 compounds. Would you clarify the text please?

As requested, this has been clarified "established drugs including pentamidine (4-4'-pentamethylendioxy)dibenzamide) as controls for any novel compounds contained therewithin."

Line 169: Since it has not been tested in GAS, it may be important to note any culture toxicity using the same concentration gradient used for the earlier test (as observed in figure 2). This would also help in indicating if pentamidine can cross the cell membrane.

This line has been edited in line with earlier requests "This aromatic diamidine is an effective antimicrobial against several eukaryotic pathogens, but efficacy against SpeB from GAS has not been established" and information on antimicrobial activity in Fig 5. Notably, other screens from this library also note no direct antimicrobial activity of pentamidine against other species of bacteria or other pathogens (refs 57-61). Regarding crossing the cell membrane, this would not be a necessary activity for pentamidine, as SpeB is inactive within the bacteria and has activity after its secretion.

Line 169: Would it strengthen the information provided to show that SpeB concentrations (only activity) is not affected in cultures using Western Blots?

This line has been edited in line with earlier requests, making the point more specific to enzymology

Line 173: Should it indicate that it is effective in vitro? Could it be referenced if the concentration found here is ideal in comparison to host cells?

As requested, this has been clarified to read “in vitro”, as well as the experimental setup, in line with other comments on needed specificity

Figure 4A-C: These would go well in the previous image, and it includes the known inhibitor E64.

In response to other requests, we have changed the flow from the inhibitor screen to the activities of pentamidine, addressing the separation here.

Figure 4D: Not able to observe any error bars, is sentence part of this graph? I think there's an additional 0 in e-009. Text in results says e-09.

As requested, we have added error bars and “009”

Line 179: It is mentioned here that 10 μM does not affect growth before addition of neutrophils. This further supports the recommendation above for lines 169 and 173. Is the 10 μM the lowest concentration that can be used?

We have not extensively tested whether lower concentrations are effective, this was similar to concentrations of various drugs typically used for in vitro assays and sufficient to have statistically significant effect

Line 180: When the manuscript says “species-specific”, does it refer to *S. aureus* also having secreted proteases? Otherwise, based on the information provided, it is not clear why *S. aureus* is a positive control if it's specific to SpeB. I guess the question would be, what is pentamidine binding to in the *S. aureus* assay? Would it be more relatable to have other Streptococcal species for comparison, ones perhaps with close or distantly related proteases?

As requested we have clarified this “Staphylococcus aureus, a bacteria included as a control that causes similar infections but lacks a SpeB homolog”

The assay could benefit from using a deletion of SpeB to determine if the decrease is due specifically to the interaction with SpeB and not general toxicity.

As requested we have added a speB-mutant strain to confirm no role for general toxicity

Figure 5: Have Figure legend match the Y-axis. Does each black dot represent the experiments for a total of 6? Not sure error bars are present.

As requested we have amended the legend to match the Y-axis, and re-rendered the image to show error bars

Discussion

Line 193: Perhaps arrange by category – parasites, fungal, bacterial, etc. Line 197-199: There's no transition here to connect with pentamidine.

As requested this has been rearranged, with additional discussion on an example of a poor inhibitor, to connect with the transition to pentamidine

Line 199: I think there is a word missing 'which enters is known'

Corrected as requested “which is known to”

Line 202: should say 'with some success'

Corrected as requested “with some success”

Line 208: Perhaps indicate that because there might be a broader target (not just *S. pyogenes*) a study of related SpeB or SpeB-like proteins in commensal organisms may be useful given the desire to minimally target the biome?

Corrected as requested with discussion "Diverse pathogens other than P. gingivalis and GAS encode cysteine protease virulence factors, leaving the possibility that pentamidine may be more broadly useful as an antimicrobial."

Line 212-218: I think here the concentrations are the point. Please refer to the above comments on host cell concentrations please.

Citation added with text correction as requested "similar concentrations as a typically required as a therapeutic or prophylactic for these diseases"

Other comments: Some future questions that perhaps could be part of the discussion include is pentamidine binding reversible / irreversible, is there known information about resistance generated in other microorganisms that could be a concern for S. pyogenes, would they observe similar results if the assay was done in mice (% survival), etc.

Added discussion as requested "Additional experiments would be required to determine the breadth of pentamidine's protease inhibitor activities, modality of inhibition, and whether resistance can evolve."

Acknowledgements

Line 227: Any information regarding the IRB?

IRB is addressed in methods, expanded on acknowledgement here "We appreciate the technical support provided by the Children's Healthcare of Atlanta and Emory University's Children's Clinical and Translational Discovery Core for whole blood and cell processing, and the human donors."

"

References

Some have the DOI and some do not. Some have the Genus and species italicized and some do not. Quick check recommended.

All references checked and formatted to ASM style

Re: Spectrum00758-25R1 (**Pentamidine inhibition of streptopain attenuates *Streptococcus pyogenes* virulence**)

Dear Dr. Christopher N. LaRock:

Your manuscript has been accepted, and I am forwarding it to the ASM production staff for publication. Your paper will first be checked to make sure all elements meet the technical requirements. ASM staff will contact you if anything needs to be revised before copyediting and production can begin. Otherwise, you will be notified when your proofs are ready to be viewed.

Sincerely,
Justin Kaspar
Editor
Microbiology Spectrum

Summary

The *edited* manuscript titled “Pentamidine inhibition of streptopain attenuates *Streptococcus pyogenes* virulence” provides information regarding hits from a library and a separate molecule, pentamidine, that have a variety of inhibitory activities against the cysteine protease SpeB in *S. pyogenes*. The manuscript argues that targeting specifically SpeB in an *in vitro* setting will help reduce the number of molecules that may have indirect effects. They hypothesize that a reduction in SpeB activity can allow the immune system to target and neutralize *S. pyogenes* more effectively. To this end, they test a compound library against purified SpeB and provide activity and modeling for 7 targets and pentamidine. The edited study provides better information regarding the compound library setup as well as explaining the activity of the selected targets. They provided an additional assay that further supports the use of pentamidine in the presence of immune cells such as neutrophils. The last section of the results can be improved to strengthen the purpose of this manuscript. All other sections were much improved in the edited manuscript. Please find below comments using the manuscript’s line numbers.

Abstract

No edits/comments

Importance

No edits comments

Introduction

No edits comments

Methods

Line 90. GAS 5448 is the wild type, correct? The $\Delta speB$ comes from this strain as well, correct?

Line 93. Add Phosphate Buffer Saline

Line 93. ‘Native SpeB’ refers to the active protein, correct?

Line 99. Add Dimethyl sulfoxide

Line 100. Add ‘were performed (or done) in PBS for screening.....

Line 106. The lead hit pentamidine was part of the compound library? I believe this gets mentioned in the results section but should be clarified here. Were there other compounds that needed to be dissolved in PBS? The switch to PBS did not lead to precipitation?

Line 111. This sentence (‘Modeling was agnostic....’) does not read well. Is it missing something? Define ptm for this section.

Line 117. Define VDW parameter.

Line 128. Define activity bins in this section.

Line 134. Add 10% FBS (*Fetal Bovine Serum*).

Line 135. Add CFU/mL; colony forming units per mL.

Line 137. Add CFU/mL

Results (Figure comments here for context)

Line 499. Figure 1 legend / Figure 1A. Relative Fluorescence Units (RFU)

Line 504. Figure 2 legend. Indicate X and Y axis. X axis says seconds, results text says after 30 minutes. I assume the run starts in seconds after the 30 minute incubation correct? Provide in the legend what the starts next to the lines means; * = I think you did this correctly in Figure 5 legend.

Line 159. Should it say (data not shown).

Line 169. As requested previously, this section would benefit from providing a reference to the SpeB amino acid region observed in the images. It was requested that a summary of interactions was placed. I do see this was done for pentamidine but not for the other seven that were kept separate. It would make the section stronger to know if say, 5/7 had interactions using the same predicted amino acids. From looking at the images some of the compounds are larger or smaller which could relate back to why diluting them is affecting the activity.

Line 509. Figure 3 legend. Add SpeB region here for reference.

Line 516 Figure 4 legend. Should state X and Y axis for reference in 4D.

Line 184-185. There is a jump between the information for 4A-C (stronger section) to 4D. You only have 1 sentence, the punchline, but I got nothing to attach it to. Why is it suggestive that it can be effective, the number does not mean much if is not compared to or given relevance. I believe there was something in the Introduction section that could assist here to reinforce perhaps.

Line 191. Again here, you provide a punchline but not prior explanation. 1-2 sentence setup between (*in vivo*. 10 μ M) might help with that. Which strains were used for the assay and which ones are controls is very important here as you will have in the image different combinations of pentamidine / neutrophils in figure 5.

Line 191 -198. This section would benefit from explaining each graph by itself. As it stands, it is a series of punchlines, and you have to go back and forth between the images to understand it. Each graph gives support to the next one and showing that losing SpeB

in the 3rd graph is the final strong point for pentamidine. I would suggest rearranging as this is your most important section.

Figure 5. Addition of the $\Delta speB$ graph is very helpful for the comparison.

Discussion

Line 214. Space missing at the end of sentence.

Line 217. As previously requested, is pentamidine the 8th molecule that was a hit in this study, or was it one of the seven molecules, or was not counted because it was a control of the library?

Line 231. Needs reference.

Line 238. Add 'microbial resistance'.